# Effects of Low-Salinity Stress on Histology and Metabolomics in the Intestine of *Fenneropenaeus chinensis*

**DOI:** 10.3390/ani14131880

**Published:** 2024-06-26

**Authors:** Caijuan Tian, Qiong Wang, Tian Gao, Huarui Sun, Jitao Li, Yuying He

**Affiliations:** 1State Key Laboratory of Mariculture Biobreeding and Sustainable Goods, Yellow Sea Fisheries Research Institute, Chinese Academy of Fishery Sciences, Qingdao 266071, China; tcaijuan@163.com (C.T.); wangqiong@ysfri.ac.cn (Q.W.); linyigaot@163.com (T.G.); sunhuarui36@163.com (H.S.); 2Jiangsu Key Laboratory of Marine Biotechnology, Jiangsu Ocean University, Lianyungang 222005, China; 3Laboratory for Marine Fisheries Science and Food Production Processes, Qingdao Marine Science and Technology Center, Qingdao 266237, China; 4College of Fisheries and Life Science, Dalian Ocean University, Dalian 116023, China

**Keywords:** shrimp intestine, morphology, metabolites, ABC transporters, salinity

## Abstract

**Simple Summary:**

Acute low-salinity stress can affect intestinal microbiota composition and differentially expressed genes in *Fenneropenaeus chinensis*. However, the effects of acute low-salinity stress on the metabolic profile of *F. chinensis* remain unclear. In this study, intestinal histological examination and untargeted metabonomic analysis of *F. chinensis* were conducted after exposure to low-salinity stress at different time points. Our results demonstrated that the intestinal morphological structure of *F. chinensis* and many differential intestinal metabolites were present in a low-salinity environment. Fourteen key metabolites were screened to predict their metabolism under stressful conditions. Furthermore, some related pathways, such as phenylalanine, tyrosine, and tryptophan biosynthesis; fatty acid and retinol metabolism; and ABC transporters, were significantly enriched, which played an important role in the low-salinity environment. This study contributes to our understanding of the molecular response mechanisms to low-salinity stress in shrimp.

**Abstract:**

Metabolomics has been used extensively to identify crucial molecules and biochemical effects induced by environmental factors. To understand the effects of acute low-salinity stress on *Fenneropenaeus chinensis*, intestinal histological examination and untargeted metabonomic analysis of *F. chinensis* were performed after exposure to a salinity of 15 ppt for 3, 7, and 14 d. The histological examination revealed that acute stress resulted in most epithelial cells rupturing, leading to the dispersion of nuclei in the intestinal lumen after 14 days. Metabolomics analysis identified numerous differentially expressed metabolites (DEMs) at different time points after exposure to low-salinity stress, in which some DEMs were steadily downregulated at the early stage of stress and then gradually upregulated. We further screened 14 overlapping DEMs, in which other DEMs decreased significantly during low-salinity stress, apart from L-palmitoylcarnitine and vitamin A, with enrichments in phenylalanine, tyrosine and tryptophan biosynthesis, fatty acid and retinol metabolism, and ABC transporters. ABC transporters exhibit significant abnormalities and play a vital role in low-salinity stress. This study provides valuable insights into the molecular mechanisms underlying the responses of *F. chinensis* to acute salinity stress.

## 1. Introduction

The Chinese shrimp (*Fenneropenaeus chinensis*) is a well-established and migratory species in aquaculture in China, primarily distributed in the coastal regions of the Yellow and Bohai Seas in northern China [1,2,3]. This species is highly valued for its nutritional composition and ability to thrive in artificial culture environments [2]. However, in the process of pond culture, continuous heavy rainfall also causes rapid changes in the salinity of pond water, which disrupts the osmotic balance in crustaceans and causes a series of physiological adaptive responses, resulting in a stress response that, if prolonged in a state of stress, leads to a decline in the immunity of the organism and an increase in the incidence of disease [4,5]. Therefore, knowing how *F. chinensis* adapts to salinity fluctuations at different time points is vital to research.

Stress plays a pivotal role in modulating the immune response of shrimp, resulting in overall immunosuppressive effects and potentially impacting metabolic activities in some cases [6,7], particularly under salinity stress. However, the mechanisms of osmoregulation and molecular response to salinity stress in *F. chinensis* are unknown. Salinity stress enhances cells’ reactive oxygen species (ROS) content and creates oxidative stress [6], ultimately affecting crustaceans. Shrimps have developed a complex antioxidant defense system to mitigate ROS-mediated oxidative damage that includes low-molecular-mass antioxidants [6]. This system is an integrated intracellular protective network that utilizes specific enzymes to catalyze the breakdown or conversion of ROS and non-enzymatic antioxidants to directly scavenge free radicals, which work together to maintain the stability of the intracellular environment and the healthy state of the cell [8]. Enzymes involved in the antioxidant defense system are glutathione-S-transferase (GST), catalase (CAT), glutathione peroxidase (GPx), superoxide dismutase (SOD), and others. Vitamin C, vitamin A, and reduced glutathione (GSH), are non-enzymatic defense systems [9]. Additional energy may help to maintain osmolarity in cells for homeostasis under stressful conditions.

Untargeted metabolomics aims to provide a holistic understanding of metabolic networks and fluctuation in metabolite levels by simultaneously tracking changes in all small-molecule metabolites [10]. Metabolomic profiling reveals the fundamental mechanisms between the host and metabolites using high-throughput analytical methods, which helps determine how metabolism is affected by different disease states linked to dysbiosis [11,12,13]. It is hoped that metabolomics can be used to identify key metabolic features by regulatory molecules related to metabolic pathways further to analyze the interaction mechanisms between hosts and metabolites.

Previous studies have demonstrated remarkable gene expression effects in *F. chinensis* in response to low-salinity stress [14]. However, it has been observed that different omics approaches provide different results, and relying only on a single omics dataset may not fully reflect the changes in various biomolecules during stress response [15,16]. In this study, liquid chromatography–mass spectrometry (LC-MS) technology was used to comprehensively investigate the changes in molecular metabolites in the intestine of *F. chinensis*, which is of great theoretical and practical significance for the development of new species for antiretroviral aquaculture, adjustment of aquaculture structure in watersheds, and development of sustainable fisheries.

## 2. Materials and Methods

### 2.1. Animals

A total of 180 healthy and standard individuals (weight 9.53 ± 1.55 g) were obtained from the Haifeng Fishery Technology Company (Changyi, China). They included randomly selected male and female shrimps above four months of age placed in a tank for seven days (salinity at 30 ppt). For other details regarding the animals, please refer to [14].

### 2.2. Low Salinity Stress Experiment and Sampling

The methods used for the low-salinity stress experiment and sampling were the same as those described in a previous study [14]. According to the results of another study [17], the mortality rate of *F. chinensis* was 100% after 72 h at a salinity of 10 ppt; whereas, the average survival rate of the shrimp at a salinity of 15 ppt was 65%. Therefore, in brief, after 7 days of acclimation, the shrimp were placed in a 71 × 50 × 43 cm (150 L) tank, then in a low-salinity environment (15 ppt) for 0 days (C0), 3 days (S3), 7 days (S7), and 14 days (S14), to represent different experimental time points. There were 9 shrimps per parallel experiment and 27 shrimps per stress time point. Intestinal samples were collected separately at different stress time points and then placed in liquid nitrogen for further metabolomic analysis; the shrimps were not fasted before sampling.

### 2.3. Histological Analyses of Intestine

In addition, 36 intestinal fragments (approximately 0.5 cm) from the shrimp midsection at different stress time points were collected and immediately fixed in 4% paraformaldehyde fixative for 24 h, and then stored in 70% ethanol for intestinal tissue section analysis. After rinsing in water, the tissues were dehydrated in a series of ethanol (70%, 80%, 90% and 100%), cleared with xylene, embedded in paraffin and cut in a microtome (Leica, RM2016, Wetzlar, Germany) at a thickness of 6 μm. After hematoxylin–eosin (HE) staining, the slides were observed and photographed using an optical microscope (Pannoramic MIDI II, Budapest, Hungary).

### 2.4. Untargeted Metabonomic Analysis

For sample preparation, 100 mg of intestinal tissue from each repeat with 27 shrimps was individually ground with liquid nitrogen, and the homogenate was resuspended in pre-chilled 80% methanol by vortexing. The samples were incubated on ice for 5 min and then were centrifuged at 15,000× *g*, 4 °C for 20 min. Some of the supernatant was diluted to a final concentration of 53% in methanol by LC-MS-grade water. Subsequently, the supernatant was clarified by centrifugation at 15,000× *g*, 4 °C for 15 min (D3024R, Scilogex, USA), which was then transformed into the LC-MS system analysis [18]. Quality control (QC) samples were prepared by mixing aliquots of all samples into one pooled sample.

Mass spectrometry conditions: The scan range was range from 100 to 1500 *m*/*z*; QExactiveTM HF-X mass spectrometer (Thermo Fisher, Dreieich, Germany) was operated in positive/negative polarity mode with SprayVoltage of 3.5 kV, Sheath gas flow rate of 35 psi, AuxGas flow rate of 10 L/min, Capillary Temp of 320 °C, S-lens RF level of 60, Aux gas heater temp of 350 °C.

The CompoundDiscoverer 3.1 (CD3.1, Thermo Fisher Scientific) was used for peak identification, data correction, and metabolite identification. The main parameters were set as follows: retention time, 0.2 min; actual mass, 5 ppm; signal intensity, 30%; signal/noise ratio, 3; and minimum intensity. The peaks were then matched using the mzCloud (https://www.mzcloud.org/ accessed on 23 May 2023), mzVault, and MassList databases to obtain accurate qualitative and quantitative results. Pearson correlation coefficients between QC samples were calculated based on the relative quantitative values of the metabolites [19].

### 2.5. Identification and Multivariate Statistical Analysis of Metabolites

Metabolites were annotated using the Kyoto Encyclopedia of Genes and Genomes (KEGG) (https://www.genome.jp/kegg/pathway.html accessed on 10 February 2018), Human Metabolome Database (HMDB) (https://hmdb.ca/metabolites accessed on 23 February 2022) and LIPIDMaps (http://www.lipidmaps.org/ accessed on 5 November 2021) databases. Metabolites with Variable Importance in Projection (VIP) [20] >1 from Partial Least Squares Discriminant Analysis (PLS-DA) [21,22] and *p* value < 0.05 (Student’s *t*-test) were considered significant metabolic perturbations between the two groups. To compare differences between groups, a discriminant analysis method using PLS-DA was used together with a permutation test to confirm that supervised learning methods to obtain classifications were not fortuitous. Clustering heat maps were plotted using the R package heatmap, and metabolite data were normalized using the z-score. Correlation analysis (Spearman’s correlation coefficient) between differentially expressed metabolites (DEMs) was performed using the Corrplot package in R. Bubble. We investigated metabolites’ function and metabolic pathways using the metaX [23], which were considered significantly enriched when *p*-values were <0.05.

## 3. Results

### 3.1. Histological Analysis of the Intestine

The morphological changes in the intestinal tissue of *F. chinensis* after low-salinity stress are shown in Figure 1. Histological observation showed that the intestinal tissue in shrimp in the control group (C0, Figure 1A) had a normal morphology, a well-developed network of villi almost filling the entire intestinal lumen, and epithelial cells with uniformly distributed columnar and cup-shaped nuclei. In S3 (Figure 1B), the changes in tissue morphology were not obvious. In S7 (Figure 1C), the epithelial cells were deformed, and vacuoles appeared. In contrast, under prolonged low-salinity stress, the S14 group (Figure 1D) displayed many intestinal alterations. For example, most epithelial cells were necrotic, and even ruptured; epithelial cells were detached from the basement membrane, and deformed, and loosely attached; nuclei were dispersed in the intestinal lumen; intestinal villi were gradually atrophied, and their height showed a tendency to decrease.

### 3.2. Identification of Metabolic Changes

A total of 1459 metabolic signals from 36 samples were detected, including 1050 and 409 metabolites in the positive and negative ion modes, respectively (Appendix A), indicating their robustness and reliability. Typical total ion chromatograms of intestinal samples were obtained from *F. chinensis* in the positive mode (Appendix A). Pearson’s correlation showed that all the QC samples were of good quality (≥0.988 (Appendix A). Principal Component Analysis (PCA) revealed some differences after low-salinity stress (Appendix A). The PLS-DA model demonstrated a significant metabolic difference between the control and experimental groups, with Q2 values of 0.94, 0.98, and 0.97 in the S3, S7, and S14 groups, respectively, and R2 values of 0.98, 1.00, and 0.99 in the S3, S7, and S14 groups, respectively (Figure 2). Taken together, these results are consistent and can be further analyzed using metabolomics. Metabolites were divided into nine main categories: lipids and lipid-like molecules (50.90%) (Appendix A). Lipid maps (Appendix A) and KEGG pathway annotations (Appendix A) were used to understand the functional properties and classifications of the identified metabolites.

### 3.3. Identification and Enrichment Analysis of DEMs

In total, 382 (183 up-regulated and 199 down-regulated), 490 (213 up-regulated and 277 down-regulated), and 546 (274 up-regulated and 272 down-regulated) DEMs were identified in S3, S7, and S14, respectively (Figure 3A). Cluster analyses of DEMs portrayed different expression profiles among the three treatment groups (Figure 3B–D). Using a Venn diagram, a total of 134 DEMs from the two ion modes were simultaneously regulated by low-salinity stress (Figure 3E,F). To further analyze the biological pathways associated with metabolic changes in *F. chinensis* under low-salinity stress, all identified DEMs were mapped to the KEGG pathway database, and the top 20 enriched pathways were subsequently analyzed (Figure 4). Significant differences in the functional and metabolic pathways were observed at different time points. For example, as also shown in Appendix A, intriguingly, in S3, we observed that acute salinity stress induced significant enrichment of 28 DEMs in seven pathways in comparison with C0, including phenylalanine, tyrosine, and tryptophan biosynthesis; galactose metabolism; inositol phosphate metabolism; carbohydrate digestion and absorption. In S7, some pathways, including ABC transporters, were enriched with metabolites. In S14, fatty acid biosynthesis and phenylalanine metabolism were significantly enriched.

### 3.4. Identification and Analysis of 14 Key DEMs

To investigate the effects of low-salinity stress on *F. chinensis*, we performed an untargeted metabolomic analysis of the shrimp intestine using an LC-MS platform. Overlapping DEMs were enriched in phenylalanine metabolism; phenylalanine, tyrosine, and tryptophan biosynthesis; ABC transporters; retinol metabolism; and fatty acid degradation (Appendix A). Among these pathways, 14 metabolites were selected as key DEMs and their information is shown in Table 1. Vitamin A and L-palmitoylcarnitine (L-Pcar) were up-regulated, whereas other DEMs were down-regulated.

Subsequently, we examined the relative changes (Figure 5A) and correlations (Figure 5B) among the 14 DEMs to further understand the metabolite changes in response to low salinity stress. The content of all these DEMs consistently decreased during low-salinity stress, apart from vitamin A and L-Pcar. Overall, the 14 DEMs were closely related, which is significant for explaining the pathogenesis of stress. For instance, Tryptophan (Trp) was more positively closely related to 6-Hydroxymelatonin (6-OHM), Succinic acid (SA), L-Kynurenine (L-Kyn), and L-Phenylalanine (L-Phe), in which L-Phe was more positively closely related to 6-OHM. Notably, Sucrose (SUC) was more positively closely related to Maltotriose (Mal) and Hippuric acid (HA). In contrast, L-Pcar and vitamin A were negatively related with the other metabolites.

## 4. Discussions

### 4.1. Low-Salinity Stress Induced Micromorphic and Metabolic Changes in Intestine

As a result of the decrease in arable land resources in developing countries, it is important to evaluate the salinity tolerance of shrimp species and their ability to grow in salinized environments [24]. As a key metabolic organ, the intestine governs energy metabolism such as glucose, lipid, and amino acid metabolism [25]. Consistent with the results of [26], compared to the control group (pH 8.3) under acute stress conditions, both low and high pH stresses disrupted the intestinal morphological structures. Previously, it was demonstrated that stress leads to intestinal structural damage, including goblet cells, and affects intestinal villus and wall integrity [27], suggesting that autoimmunity might be disrupted by stress in shrimp. In our study, intestinal epithelial cells were stripped from the basement membrane under low-salinity stress for 14 days, which is consistent with the result of Wang [28] and Duan et al. [29] under environmental stress. Consequently, episodes of tissue injury and incomplete healing of the intestinal epithelium are prerequisites for immune reactivation [30].

Although previous studies have examined the adaptive transcriptomic response of *F. chinensis* to low salinity [3,14,31,32], we used LC-MS-based metabolomics to determine its rapid adaptive and acute stimulatory response to low-salinity stress. Our study indicated that some DEMs were steadily down-regulated at the early stage of stress in S3 and S7, and these DEMs were more than the up-regulated metabolites during salinity stress, suggesting that acute low salinity exposure induced a drastic change in the metabolite content of the intestine. In a previous study, it led to the general depression of a large number of metabolites under stress exposure [33,34]. With the extension of stress time, *F. chinensis* gradually adapted to the saline environment through its regulation, and the number of up-regulated DEMs was approximately equal to the number of down-regulated metabolites in S14. These results align with our findings from transcriptome analyses of low-salinity stress [14]. Additionally, we screened 14 overlapping DEMs that regulated some KEGG pathways. Metabolic abnormalities in these pathways are the main causes of acute low-salinity stress, the effects of which on *F. chinensis* in low-salinity environments are elucidated below.

### 4.2. Effects of Low-Salinity Stress on Fatty Acid and Retinol Metabolism

Among the major nutritional components, lipids represent a complex group of biomolecules for the storage of energy and signal transduction, which maintain homeostasis in the presence of environmental stresses, and many fatty acids from lipid metabolism are essential for responding to ambient salinity change in animals [35,36,37]. A study demonstrated that the fatty acids metabolism plays a vital role in osmoregulation under long-term low-salinity stress [38]. Based on the results of this study, we conclude that fatty acids metabolism could be activated under acute low-salinity stress for 14 days.

Our study indicates that the role of L-Pcar is to regulate fatty acid degradation and metabolic pathways and increases content levels of this metabolite under salinity stress. The expression of up-regulated L-Pcar can increase the levels of palmitate and triacylglycerides and decrease the levels of free cholesterol, which can modulate signal transduction pathways [39]. This suggests that shrimp require extra energy from fatty acids to adapt and survive under low-salinity stress [32,39].

Enrichment analysis showed that retinol metabolism is also involved in low-salinity stress. As an essential nutrient, animals’ initial vitamin A status at the time of infection may be an important factor in their ability to respond to intestinal infection [40]. In the present study, *F. chinensis* enhanced innate and adaptive immunity by regulating retinol metabolism in the intestine and increasing the levels of vitamin A induced salinity stress to adjust to the environment. Hence, the upregulation of these DEMs in this study indicated their importance in the response of *F. chinensis* to acute low-salinity stress.

### 4.3. Effects of Low-Salinity Stress on Phenylalanine, Tyrosine, and Tryptophan Biosynthesis

Amino acid metabolism, one of the most enriched metabolic pathways, regulates essential cellular functions, such as complementary glucose and fatty acid-driven bioenergetic pathways, and can be metabolized to energy substances or used as signaling molecules to regulate cellular homeostasis [41,42]. Some amino acids that change in concentration during salinity acclimation play important roles in cell volume regulation [43]. These represented down-regulated metabolites of HA, Phenylpyruvic acid (PPA), Phenylacetylglycine (PAG), L/D-Phe, Trp, 6-OHM, and L-KYN. Again, these metabolites suggest that acute salinity induces disturbances in osmolality and energy metabolism compared to the metabolic changes induced by reduced salinity [44]. Previously, the concentrations of amino acids were reduced to balance the osmolarity under reduced seawater salinity [44]. While phenylalanine is an essential amino acid (or it cannot be synthesized by humans), Trp is considered a conditionally indispensable amino acid because it can be synthesized by the hydroxylation of phenylalanine by the enzyme phenylalanine hydroxylase [45]. It easily induces the biosynthesis pathways (phenylalanine, tyrosine, and tryptophan biosynthesis) in adverse conditions, which involves the regulation of energy metabolism [45,46,47]. Similar results were also observed in the present study, where the expression of eight metabolites, including phenylalanine and Trp significantly decreased the regulation of amino acid metabolism under low-salinity conditions. Trp was consistently suppressed by exposure to salinity stress, possibly reducing inflammatory factors and disease and enhancing immune function. In addition, PPA, PAG, SA, and HA play a vital role in adapting to ambient stress [48,49,50]. Kynurenine and 6-OHM is also an intimate relationship between tryptophan metabolism disorder and diseases [51,52,53,54,55]. SA is an endogenous isomer of methylmalonic acid that is widely distributed in human serum and urine [56]. SA plays a vital role in various metabolic pathways, including phenylalanine, tyrosine, and tryptophan metabolism and biosynthesis. It can prompt an immune response, resist ammonia stress, and suppress pathogenic infection in shrimp [50]; however, down regulation of SA resulted in low-salinity-stress-induced impairment of the immune system of *F. chinensis*. 6-OHM is an enzymatic metabolite and degradant of melatonin and is thought to be a direct free radical scavenger that provides tissue protection [55]. The use of 6-OHM enhances antioxidant and anti-inflammatory activity, which is the body’s defense mechanism against Alzheimer’s disease [57]. These results strongly suggested that DEMs related to amino acid metabolism are actively involved in energy generation processes for salinity acclimation in the intestine of *F. chinensis* at low salinity concentrations.

### 4.4. Effects of Low-Salinity Stress on ABC Transporters

Salinity stress also leads to osmotic imbalance, oxidative stress, and ionic disturbance, and even an increase in the amount and serious imbalance of ROS within animal cells [58]. Acute salinity induces disturbances in energy metabolism compared to the metabolic changes induced by reduced salinity and may interfere with normal energy intake and expenditure [44]. During this process, aquatic animals are forced to alter the expression of various enzymes and transporters to generate metabolic activities [59,60]. For instance, ABC transporters, a 48-member superfamily of membrane proteins, actively transport various biological substrates across lipid membranes (including exogenous substances) and are involved in the absorption of small molecules [61]. 

Previous studies have shown that ABC transporters play key roles in the physiology and development of plants’ responses to abiotic stresses [62,63,64], but few studies are concentrated on crustaceans. In the present study, some ion transporter metabolites were inhibited after *F. chinensis* was subjected to low-salinity stress, which reduced the intestinal absorption of ions, the ion loss, and ATP consumption during stress. DEMs, including Mal, SUC, and Biotin, are involved in inducing ABC transporters under low-salinity stress. Mal is largely resistant to the action of the body but is readily hydrolyzed to glucose by homogenates of the small intestinal mucosa [65]. As a metabolite, SUC plays a crucial role in development, stress response, and yield by promoting the growth and synthesis of essential compounds [66]. Biotin (Vitamin B7), a water-soluble B-vitamin, is an essential micronutrient for cellular functions, and its receptors are overexpressed in certain cancers [67]. In conclusion, ABC transporters play a key role in *F. chinensis* in response to low-salinity stress.

## 5. Conclusions

This is the first study to demonstrate that low-salinity stress causes apparent histological changes in the intestine of *F. chinensis* after 14 days of exposure. Further, untargeted metabolic profiling indicated that there were 14 overlapping DEMs enriched in phenylalanine, tyrosine, and tryptophan biosynthesis, ABC transporters, fatty acid, and retinol metabolism, in which the content of some DEMs was decreased, while the corresponding metabolites also increased to underlie the rapid adaptive and acute stimulatory responses to low-salinity stress. Key metabolic pathways are involved in the regulation of adverse effects caused by low-salinity stress in the intestine of *F. chinensis*. These findings provide new insights into the regulatory mechanisms of the intestine in *F. chinensis* and essential information for the breeding *F. chinensis* tolerant to low-salinity stress.

## Figures and Tables

**Figure 1 animals-14-01880-f001:**
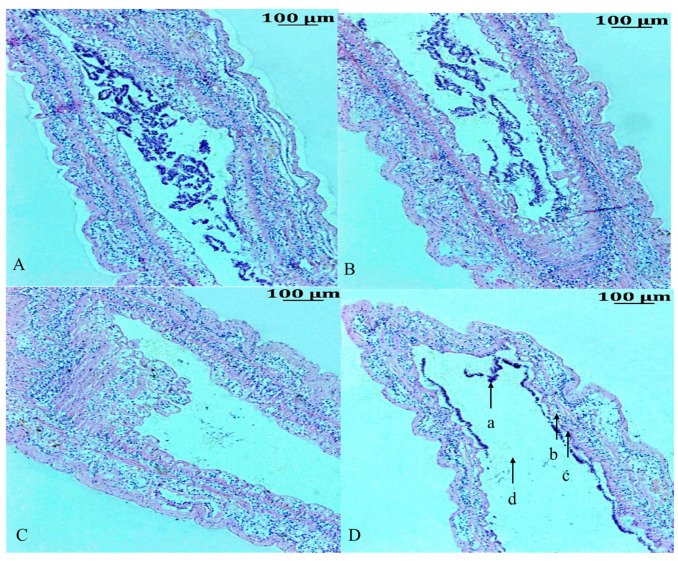
Intestine histological morphology of *F. chinensis* under low-salinity stress. (**A**) Intestine structure of C0, Χ400; (**B**) intestine structure of S3, Χ400; (**C**) intestine structure of S7, Χ400; (**D**) intestine structure of S14, Χ400. (a) Nuclei, (b) epithelium, (c) brush border, (d) lumen, rule: 100 μm.

**Figure 2 animals-14-01880-f002:**
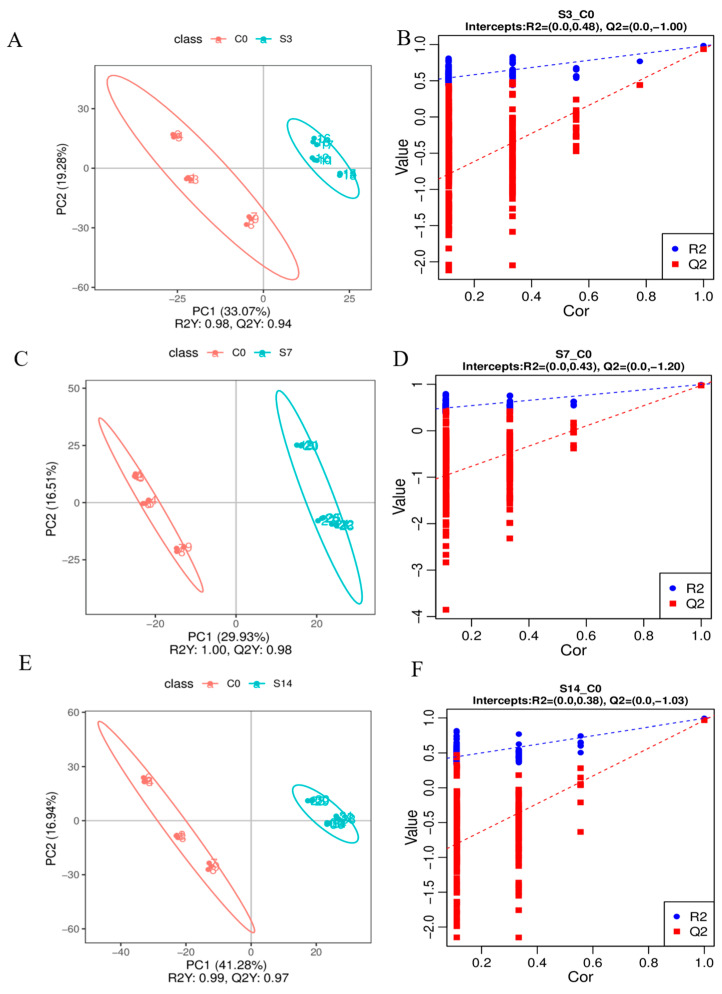
Multivariate modeling of LC-MS data in response to low-salinity stress in *F. chinensis*. (**A**) PCA score plot and (**B**) validation PLS-DA score plot in S3; (**C**) PCA score plot and (**D**) validation PLS-DA score plot in S7; (**E**) PCA score plot and (**F**) validation PLS-DA score plot in S14.

**Figure 3 animals-14-01880-f003:**
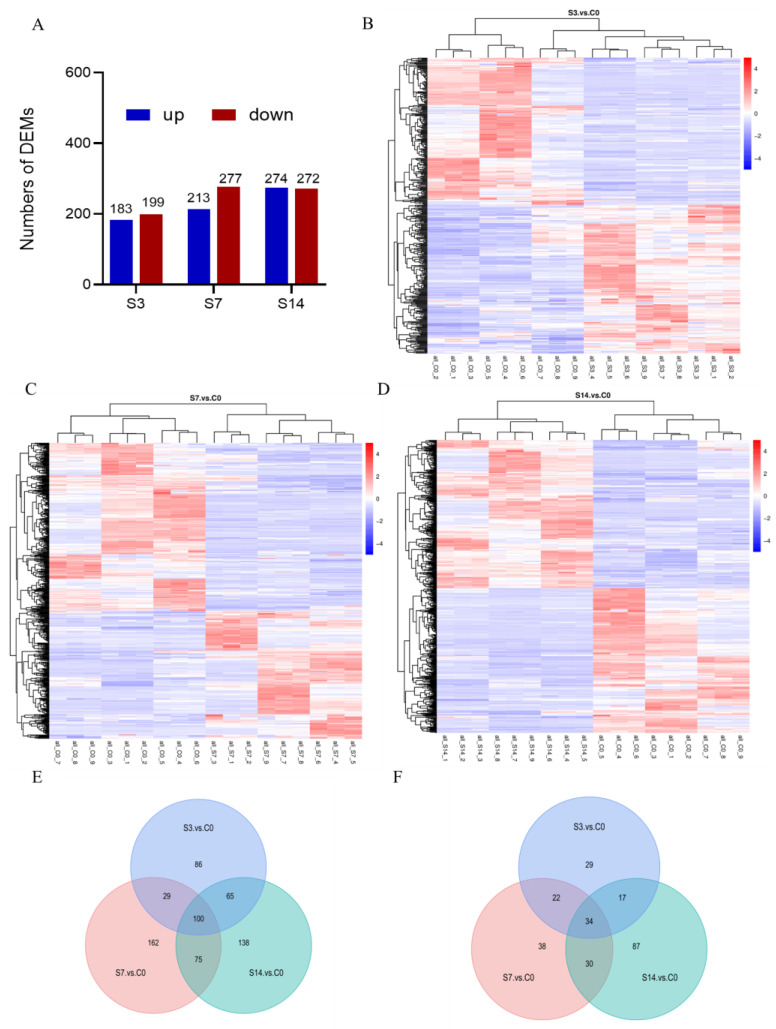
Information of intestine differentially expressed metabolites (DEMs) of *F. chinensis* under low-salinity stress. (**A**) numbers of DEMs in S3, S7, and S14 vs. C0. (**B**) DEMs are expressed by a heatmap in S3. (**C**) DEMs are expressed by a heatmap in S7. (**D**) DEMs are expressed by a heatmap in S14. DEMs indicated by a Venn diagram in the positive (**E**) and negative (**F**) iron mode.

**Figure 4 animals-14-01880-f004:**
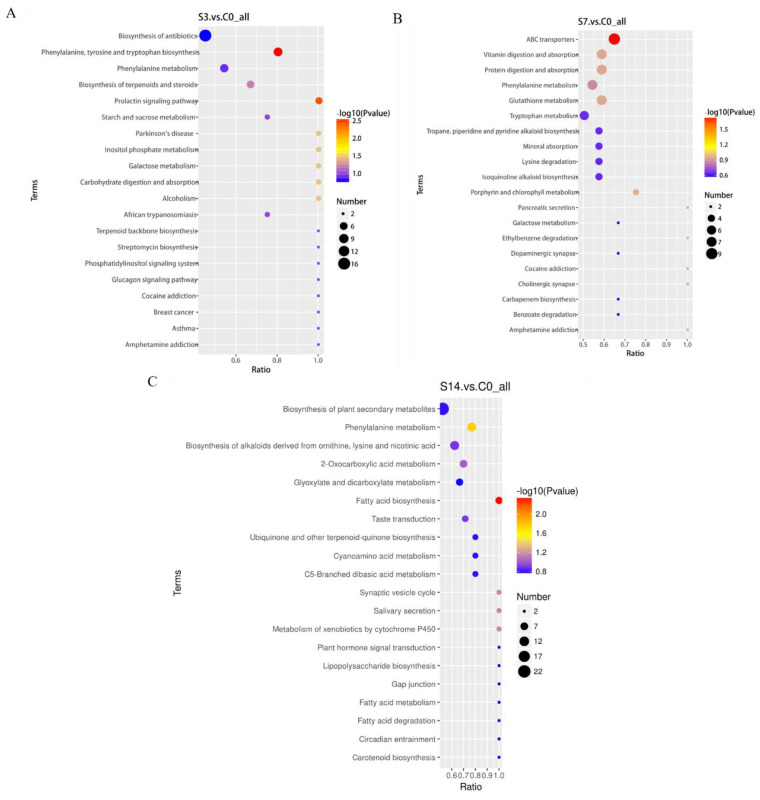
The top 20 KEGG enrichment analysis of DEMs in response to low-salinity stress. (**A**) enrichment analysis of DEMs in S3 vs. C0. (**B**) enrichment analysis of DEMs in S7 vs. C0. (**C**) enrichment analysis of DEMs in S14 vs. C0.

**Figure 5 animals-14-01880-f005:**
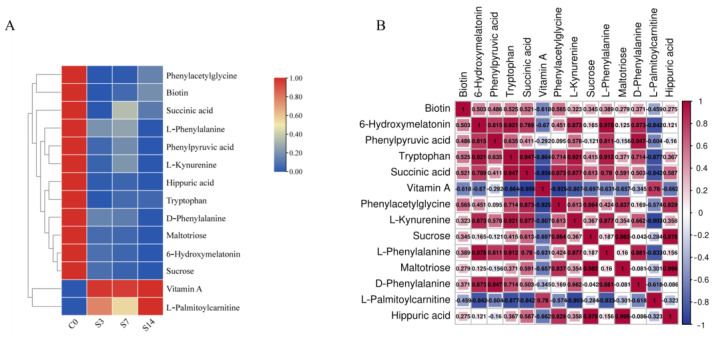
Information of 14 key DEMs in the intestine of *F. chinensis* in response to low-salinity stress. (**A**) Heatmap and (**B**) Spearman’s correlation analysis of DEMs. Numbers represent the correlation coefficient values between different DEMs; red and blue indicate positive and negative correlation.

**Table 1 animals-14-01880-t001:** Metabolite differential screening results.

Mode	Name	Up/Down	Formula	RT [min]	*m*/*z*	*p* Value in S3, S7, S14		
Pos	Hippuric acid (HA)	Down	C_9_H_9_NO_3_	5.131	180.0657	0.008725318	0.016019771	0.005866128
	Phenylpyruvic acid (PPA)	Down	C_9_H_8_O_3_	5.035	165.0547	2.66021 × 10^−5^	0.00059833	1.03938 × 10^−5^
	L-Phenylalanine (L-Phe)	Down	C_9_H_11_NO_2_	5.012	166.0865	0.002437022	0.005238281	8.70015 × 10^−5^
	D-Phenylalanine (D-Phe)	Down	C_9_H_11_NO_2_	10.229	166.0868	0.00534596	0.003718651	0.000552025
	6-Hydroxymelatonin (6-OHM)	Down	C_13_H_16_N_2_O_3_	5.442	249.1245	1.4535 × 10^−6^	3.12169 × 10^−7^	1.54119 × 10^−7^
	L-Kynurenine (L-Kyn)	Down	C_10_H_12_N_2_O_3_	5.272	209.0925	0.001332862	0.010433043	0.0011305
	Vitamin A	Up	C_20_H_30_O	9.36	287.2369	0.000773157	0.001374898	0.004364512
	L-Palmitoylcarnitine (L-Pcar)	Up	C_23_H_45_NO_4_	10.234	400.3429	0.007961563	0.010861824	0.001583441
Neg	Succinic acid (SA)	Down	C_4_H_6_O_4_	2.469	117.0182	0.000417082	0.023997658	0.002328271
	Phenylacetylglycine (PAG)	Down	C_10_H_11_NO_3_	5.606	192.066	0.000971555	0.0008622	0.005389868
	Tryptophan (Trp)	Down	C_11_H_12_N_2_O_2_	5.219	203.0823	0.000142969	0.000869403	0.000486337
	Maltotriose (Mal)	Down	C_18_H_32_O_16_	1.385	539.1381	0.004708651	0.007046087	0.001962056
	Sucrose (SUC)	Down	C_12_H_22_O_11_	1.364	341.1101	0.002328382	0.001675135	0.000357143
	Biotin	Down	C_10_H_16_N_2_O_3_S	5.394	243.0809	1.13455 × 10^−6^	3.62306 × 10^−5^	5.47528 × 10^−5^

## Data Availability

Data will be provided on request.

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
