# Peer review of "Effects of Low-Salinity Stress on Histology and Metabolomics in the Intestine of Fenneropenaeus chinensis"

_animals, 2024, doi:10.3390/ani14131880_

Round 1

Reviewer 1 Report

Comments and Suggestions for Authors

The manuscript entitled "Effects of low salinity stress on histology and metabolomics in the intestine of Fenneropenaeus chinensis" provides new information on the histology and metabolomics in the intestine of the shrimp Fenneropenaeus chinensis after exposure to 15 ppt salinity for 3, 7 and 14 days. It was shown that the majority of epithelial cells were ruptured, resulting in the dispersion of nuclei in the intestinal lumen after 14 days. Numerous differential metabolites were identified in the treatment groups. The results obtained are interesting and novel from the point of view of explaining the mechanisms of the organism's response to stress factors such as changes in salinity.

The introduction is quite well written and provides comprehensive information on the importance of studying the influence of salinity fluctuations on shrimps. The experiment is competently organised. Detailed descriptions of the experiment, sampling, histological and metabolomic analyses are given. The conclusion section summarises information on the key metabolic pathways involved in the response of F. chinensis to low salinity stress.

Minor comments:

1.         Lines 94-95. Please check this sentence. What does the number 715043 mean?

2.         Lines 94-95. You write "Shrimp were placed in the tank with low salinity environment (15 ppt) ... for 0 days (C0, salinity 30 ppt) ...". There is an inconsistency in this sentence, please rephrase.

3.         Line 101. Please state the number of animals from which material was collected for histological analysis.

4.         Line 109. Please indicate the number of animals from which material was collected for metabolomic analyses.

5.         Lines 169, 181, 184 et seq. Please check the links.

Author Response

Dear reviewers and editor,

Thank you for your constructive comments and suggestions. Your comments are very helpful to improve the quality of the manuscript. Words in red are the changes we have made in the manuscript. Now I response the reviewers’ comments with a point by point and highlight the changes in revised manuscript. We sincerely hope that our responses and modifications are acceptable for publication.

Reviewer 1:

Minor comments:

  1. Lines 94-95. Please check this sentence. What does the number 715043 mean?

Reply: I was really sorry for my careless mistakes. Thank you for your reminder. The number 715043 mean 71*50*43, representing the size of tank in the experiment. It has been modified in line 96.

  1. Lines 94-95. You write "Shrimp were placed in the tank with low salinity environment (15 ppt) ... for 0 days (C0, salinity 30 ppt) ...". There is an inconsistency in this sentence, please rephrase.

Reply: Thank you for your suggestion, I have rephrased in lines 95-98. The shrimp were placed in the tank of 71*50*43 cm (150L), then low salinity environment (15 ppt) for 0 days (C0), 3 days (S3), 7 days (S7), and 14 days (S14), respectively, to represent different experimental time points.

  1. Line 101. Please state the number of animals from which material was collected for histological analysis.

Reply: Thank you for your careful review, the number of all animals were listed at lines 102-103 of Section 2.3. 36 intestinal fragments (approximately 0.5 cm long) from the shrimp's midsection at different stress times were taken.

  1. Line 109. Please indicate the number of animals from which material was collected for metabolomic analyses.

Reply: Thank you for your careful review, the number of all animals were listed at lines 110-111 of Section 2.1. For sample preparation, 100 mg of intestinal tissues from each repeat with 27 shrimps were individually grounded with liquid nitrogen.

  1. Lines 169, 181, 184 et seq. Please check the links.

Reply: I apologize for my carelessness. For lines 169,181,184, and others, they have been corrected in lines 146, 147, 149, 150, 152, 171, 183, 184, 186, 189, 212, and line 215.

Reviewer 2 Report

Comments and Suggestions for Authors

The objective of the study is to investigate the impacts of acute low-salinity stress on Fenneropenaeus chinensis by conducting intestinal histological examinations and untargeted metabonomic analyses following exposure to salinity levels of 15 ppt for durations of 3, 7, and 14 days.

The main concerns in this manuscript are:

  1. The manuscript (MS) needs to provide clearer previous evidence to support the hypothesis. There is a deficiency in providing clear  previous evidence to substantiate the hypothesis posited within the manuscript. Strengthening the foundational support through a comprehensive review of existing literature and relevant studies would enhance the credibility and validity of the research.
  2. Clarify the experimental setting. Section 2.2 and Line 96 mention "treatment group", but only one treatment.
  3. .- Data about survival and growth would be  an interesting complement to this study. Incorporating data pertaining to survival rates and growth parameters would significantly enrich the scope and depth of the study.

Minor comments:

There are some problems with the references: "Error! Reference source not found"

Author Response

Dear reviewers and editor,

Thank you for your constructive comments and suggestions. Your comments are very helpful to improve the quality of the manuscript. Words in red are the changes we have made in the manuscript. Now I response the reviewers’ comments with a point by point and highlight the changes in revised manuscript. We sincerely hope that our responses and modifications are acceptable for publication.

Reviewer 2:

Main concerns:

1.The manuscript (MS) needs to provide clearer previous evidence to support the hypothesis. There is a deficiency in providing clear previous evidence to substantiate the hypothesis posited within the manuscript. Strengthening the foundational support through a comprehensive review of existing literature and relevant studies would enhance the credibility and validity of the research.

Reply: Thank you for your careful review, I have provided some previous evidence to my hypothesis. I added more references into the Discussions at lines 239-241, 250-251, 269-271, 287-289, 291-294, 302-311, 331-336. For example: In our study, the intestinal epithelial cells were stripped from the basement membrane in low salinity stress for 14 d, which consistent with Wang [28] and Duan et al. [29] under environmental stress. The expression of up-regulated L-Pcar could increase the levels of palmitate and triacylglycerides and decreases the levels of free cholesterol, which can modulate signal transduction pathways [39]. They represented down-regulated metabolites of HA, PPA, PAG, L/D-Phe, Trp, 6-OHM, L-KYN. Again, these metabolites suggest that acute salinity induces disturbances in osmolality and energy metabolism compared to metabolic changes induced by reduced salinity [45]. While phenylalanine is an essential amino acid (or it cannot besynthesized by humans), Try is considered a conditionally indispensable amino acid, because it can be synthe-sized by the hydroxylation of phenylalanine by the enzyme phenylalaninehydroxylase [46]. SA is the endogenous isomer of MMA, which is widely distributed in human serum and urine [57]. SA played a vital role in a variety of metabolic pathways that phenylalanine, tyrosine and tryptophan metabolism and biosynthesis in this study. It could prompt immune response, resistance to ammonia stress, suppress pathogenic infection of shrimp [51], but down regulation of SA showed low-salinity stress induced impairment of the immune system of F. chinensis. 6-OHM is a enzymatic metabolites and degradant of melatonin and also supposed to be a direct free radical scavenger to providing tissue protection [56]. Using of 6-OHM could enhance the effects of antioxidant and anti-inflammatory activity, which is the body’s own defence mechanism to combat Alz-heimer’s disease [58]. Mal, was largely resistant to the action of the body but was readily hydrolysed to glu-cose by homogenates of small intestinal mucosa [66]. SUC, as a metabolite, plays a crucial role in development, stress response and yield formation, promoting the growth and synthesis of essential compounds [67]. Biotin (Vitamin B7), as a water soluble B-vitamin, is an essential micronutrient for cellular functions, receptors of which are overexpressed in certain cancers [68].

  1. Wang, Z., Zhou, J., Li, J., et al. A New Insight Into the Intestine of Pacific White Shrimp: Regulation of Intestinal Homeostasis and Regeneration in Litopenaeus Vannamei During Temperature Fluctuation. Comparative Biochemistry and Physiology Part D: Genomics and Proteomics, 2020, 35: 100687.
  2. Duan, Y.F, Dong, H.B., Wang, Yun; Li, Hua; Liu, Qingsong; Zhang, Yue; Zhang, Jiasong, Intestine oxidative stress and immune response to sulfide stress in Pacific white shrimp Litopenaeus vannamei, Fish & Shellfish Immunology, 2017, 63: 201-207.
  3. Nałęcz, K.A., Szczepankowska, D., Czeredys, M., et al. Palmitoylcarnitine Regulates Estrification of Lipids and Promotes Palmitoylation of Gap-43. Febs Lett., 2007, 581(21): 3950-3954.
  4. Wu, H.F., Zhang, X.Y., Wang, Q., et al. A Metabolomic Investigation On Arsenic-Induced Toxicological Effects in the Clam Ruditapes Philippinarum Under Different Salinities. Ecotoxicol. Environ. Saf., 2013, 90: 1-6.
  5. Dai, X.H., Liu, M.Z., Xu, S.Y., et al. Metabolomics Profile of Plasma in Acute Diquat-Poisoned Patients Using Gas Chromatography-Mass Spectrometry. Food. Chem. Toxicol., 2023, 176: 113765.
  6. Duan, Y.F., Wang, Y., Zhang, J.S., et al. Dietary Effects of Succinic Acid On the Growth, Digestive Enzymes, Immune Response and Resistance to Ammonia Stress of Litopenaeus Vannamei. Fish Shellfish Immunol., 2018, 78: 10-17.
  7. Sakano, K., Oikawa, S., Hiraku, Y., et al. Oxidative Dna Damage Induced by a Melatonin Metabolite, 6-Hydroxymelatonin, Via a Unique Non-O-Quinone Type of Redox Cycle. Biochem. Pharmacol., 2004, 68(9): 1869-1878.
  8. Ma, X.; Zou, Y.; Tang, Y.; Wang, D.; Zhou, W.; Yu, S.; Qiu, L. High-throughput analysis of total homocysteine and methylmalonic acid with the efficiency to separate succinic acid in serum and urine via liquid chromatography tandem mass spectrometry. Journal of Chromatography B, 2022, 1193: 123135.
  9. D. S. Maharaj; H. Maharaj; E. M. Antunes; D. M. Maree; T. Nyokong; Glass, B. D.; Daya, S. 6-Hydroxymelatonin protects against cyanide induced oxidative stress in rat brain homogenates. Journal of Chemical Neuroanatomy, 2003.
  10. Messer, M., Kerry, K.R. Intestinal Digestion of Maltotriose in Man. Biochimica Et Biophysica Acta (Bba) - Enzymology, 1967, 132(2): 432-443.
  11. Rompicharla, S.V.K., Kumari, P., Bhatt, H., et al. Biotin Functionalized Pegylated Poly (Amidoamine) Dendrimer Conjugate for Active Targeting of Paclitaxel in Cancer. Int. J. Pharm., 2019, 557: 329-341.
  12. Zhang, N.X., Yang, Y.P., Li, C.N., et al. Based On 1H Nmr and Lc-Ms Metabolomics Reveals Biomarkers with Neuroprotective Effects in Multi-Parts Ginseng Powder. Arab. J. Chem., 2023, 16(7): 104840.

2.Clarify the experimental setting. Section 2.2 and Line 96 mention "treatment group", but only one treatment.

Reply: Thank you for your careful review, it has been described that 9 shrimps per parallel and 27 shrimps per stress time points in line 98.

  1. Data about survival and growth would be an interesting complement to this study. Incorporating data pertaining to survival rates and growth parameters would significantly enrich the scope and depth of the study.

Reply: Thank you for your valuable suggestion. According to the results of a previous study [17], the mortality rate of F. chinensis was 100% after 72 h at salinity 10, while the average survival rate of shrimp at salinity 15 was 65%, which can support our viewpoint and has been added in the lines 93-95. However, on growth parameters, since we used adult shrimp with a size of 9.53 ± 1.55 g, they did not show a significant change in growth rate at 2 weeks, and we therefore did not include data on growth parameters. In the future study, we will perform related data for deeply enrich the study.

Zhou Y.J. Effects of saline and alkaline stress on growth, reproduction and immune function of Fenneropenaeus Chinensis [D]. Master's Thesis, Dalian Ocean University, Liaoning, China, 2023.

Minor comments:

There are some problems with the references: "Error! Reference source not found"

Reply: I apologize for my carelessness, they have been corrected in lines 146, 147, 149, 150, 152, 171, 183, 184, 186, 189, 212, and line 215.

Reviewer 3 Report

Comments and Suggestions for Authors

This study investigated the effects of acute low salinity stress on Fenneropenaeus chinensis, as well as its implications on intestinal histopathology and metabolic analysis after exposure to salinity of 15 ppt for 3, 7 and 14 days. Which highlights the importance of the study.

Overall, the study is relevant to the production of non-conventional shrimp species in aquaculture. Which highlights your interest in publishing in Animals. The introduction is well written and addresses the most important aspects of the study. The materials and methods are ok, as are the results and discussion.

Despite presenting technical and scientific quality, there are some points that must be significantly improved before recommending it for Animals.

My main comments.

Line 39: Avoid keywords that are present in the title of the manuscript.

Line 90: Change "track" to "tank".

Line 142: Only the qualitative description of histological lesions may not be as effective in inferring the damage caused by salt stress. I suggest that it is interesting to include some semi-quantitative analysis methodology, such as histopathological alteration index, or mean assessment values.

I suggest you read:

Schwaiger, J., Wanke, R., Adam, S., Pawert, M., Honnen, ¨ W., Triebskorn, R., 1997. The use of histopathological indicators to evaluate contaminant-related stress in fish. J. Aquat. Ecosystem. Stress Recovery 6, 75–86. https://doi.org/10.1023/A:1008212000208.

Poleksic, V., Mitrovic-Tutundzic, V., 1994. Fish gills as a monitor of sublethal and chronic effects of pollution. In: Muller, R., Lloyd, R. (Eds.), Sublethal and Chronic Effects of Pollutants on Freshwater Fish. Fishing News Books, Oxford, UK, pp. 339–352.

Figure 1. The dpi of the figure must be adjusted to improve sharpness. I suggest that the images be captured on a larger scale, as it is not possible to clearly identify the injuries described in the results. Furthermore, I suggest that the images are better edited.

Author Response

Dear reviewers and editor,

Thank you for your constructive comments and suggestions. Your comments are very helpful to improve the quality of the manuscript. Words in red are the changes we have made in the manuscript. Now I response the reviewers’ comments with a point by point and highlight the changes in revised manuscript. We sincerely hope that our responses and modifications are acceptable for publication.

Reviewer 3:

main comments.

Line 39: Avoid keywords that are present in the title of the manuscript.

Reply: Thank you for your suggestion, it has been corrected in line 38. Keywords: Shrimp intestine; morphology; metabolites; ABC transporters; salinity

Line 90: Change "track" to "tank".

Reply: Thank you for your careful checks, it has been corrected in line 89.

Line 142: Only the qualitative description of histological lesions may not be as effective in inferring the damage caused by salt stress. I suggest that it is interesting to include some semi-quantitative analysis methodology, such as histopathological alteration index, or mean assessment values.

I suggest you read:

Schwaiger, J., Wanke, R., Adam, S., Pawert, M., Honnen, ¨ W., Triebskorn, R., 1997. The use of histopathological indicators to evaluate contaminant-related stress in fish. J. Aquat. Ecosystem. Stress Recovery 6, 75–86. https://doi.org/10.1023/A:1008212000208.

Poleksic, V., Mitrovic-Tutundzic, V., 1994. Fish gills as a monitor of sublethal and chronic effects of pollution. In: Muller, R., Lloyd, R. (Eds.), Sublethal and Chronic Effects of Pollutants on Freshwater Fish. Fishing News Books, Oxford, UK, pp. 339–352.

Reply: Thank you for your comments. In this study, the histologic section was aimed at revealing the damage of low salinity stress on intestine. However, the revealing of molecular mechanism for the low salinity stress on shrimp is the key target of this paper. We focus on the metabolomic changes of shrimp in response to low salinity stress. In the future study, we will perform deep analysis for thoroughly understand the tissue damage degree under low salinity stress. Thank you very much for this constructive suggestion again.

Figure 1. The dpi of the figure must be adjusted to improve sharpness. I suggest that the images be captured on a larger scale, as it is not possible to clearly identify the injuries described in the results. Furthermore, I suggest that the images are better edited.

Reply: Thank you for your suggestion, your advice has been a great help for me! I've tried my best to re-edit the intestinal tissue picture in Figure 1. The dpi of the figure has been adjusted, and the images have been captured on a larger scale.

Round 2

Reviewer 3 Report

Comments and Suggestions for Authors

Dear editor,

Thank you for the opportunity. All my comments and suggestions were answered.

My recommendation: accept.